# Electrostatic solution of massless quenches in Luttinger liquids

Paola Ruggiero[1*], Pasquale Calabrese[2,3], Thierry Giamarchi[4] and Laura Foini[5]

**1** King's College London, Strand, WC2R 2LS London, United Kingdom
**2** SISSA and INFN, Sezione di Trieste, via Bonomea 265, I-34136, Trieste, Italy
**3** International Centre for Theoretical Physics (ICTP), I-34151, Trieste, Italy
**4** Department of Quantum Matter Physics, University of Geneva,
24 Quai Ernest-Ansermet, CH-1211 Geneva, Switzerland
**5** Université Paris-Saclay, CNRS, CEA, Institut de Physique Théorique,
91191, Gif-sur-Yvette, France

★ paola.ruggiero@kcl.ac.uk

## Abstract

The study of non-equilibrium dynamics of many-body systems after a quantum quench received a considerable boost and a deep theoretical understanding from the path integral formulation in imaginary time. However, the celebrated problem of a quench in the Luttinger parameter of a one dimensional quantum critical system (massless quench) has so far only been solved in the real-time Heisenberg picture. In order to bridge this theoretical gap and to understand on the same ground massive and massless quenches, we study the problem of a gaussian field characterized by a coupling parameter $K$ within a strip and a different one $K_0$ in the remaining two semi-infinite planes. We give a fully analytical solution using the electrostatic analogy with the problem of a dielectric material within a strip surrounded by an infinite medium of different dielectric constant, and exploiting the method of charge images. After analytic continuation, this solution allows us to obtain all the correlation functions after the quench within a path integral approach in imaginary time, thus recovering and generalizing the results in real time. Furthermore, this imaginary-time approach establishes a remarkable connection between the quench and the famous problem of the conductivity of a Tomonaga-Luttinger liquid coupled to two semi-infinite leads: the two are in fact related by a rotation of the spacetime coordinates.



# 1   Introduction

The gaussian free field is arguably the simplest field theory. Its euclidean action in 2D reads

$$S = \frac{1}{2\pi K} \int \mathrm{d}y \int \mathrm{d}z \left[ (\partial_y \phi)^2 + (\partial_z \phi)^2 \right], \tag{1}$$

where $\phi$ is a scalar field, and $K$ a (uniform) parameter related to the compactification radius of the bosonic field. The gaussian free field can be solved by a variety of methods. In fact, using that the theory is quadratic, all correlation functions are determined in terms of its propagator only. Moreover, one can also exploit the fact that the theory is conformally invariant, which allows one to use all the powerful methods of conformal field theories (CFTs) [1].

As easy as it is, it found nonetheless highly non-trivial applications in many-body quantum physics. In 1D, in particular, it captures all the universal properties of interacting fermions and bosons by virtue of the Tomonaga-Luttinger liquid (TLL) paradigm [2–4], in which case $K$ is referred to as Luttinger parameter and encodes the entire information about the interaction strength among particles. In this case, the action (1) enters in the path integral formulation of equilibrium problems, namely in the study of TLLs at zero and finite temperature. Moreover, the non-equilibrium properties of the same class of systems can be understood via a path integral formulation on a different geometry [5, 6] (see Sec. 2 below).

More recently, various inhomogeneous generalisations of (1) were studied in connection to inhomogeneous and time-dependent problems in TLLs, both in- [7–13] and out-of-equilibrium [8, 14–24], mainly aiming at describing gases in traps and their non-equilibrium dynamics (see Section V in Ref. [25] for a more comprehensive review of the literature). However, if we consider an inhomogeneous Luttinger parameter $K \to K(y,z)$, conformal invariance is generically lost and, while the theory still remains quadratic, its propagator has generically to be determined numerically, as done, for example, in Ref. [11], but in some cases it can be treated analytically, as e.g. in Refs. [38, 53, 54]. We mention that TLLs are also characterised by a second parameter $u$, known as sound velocity. However, an inhomogeneous $u \to u(y,z)$ can be reabsorbed in a change of metric in Eq. (1), as explicitly worked out, e.g., in [7, 8, 11].

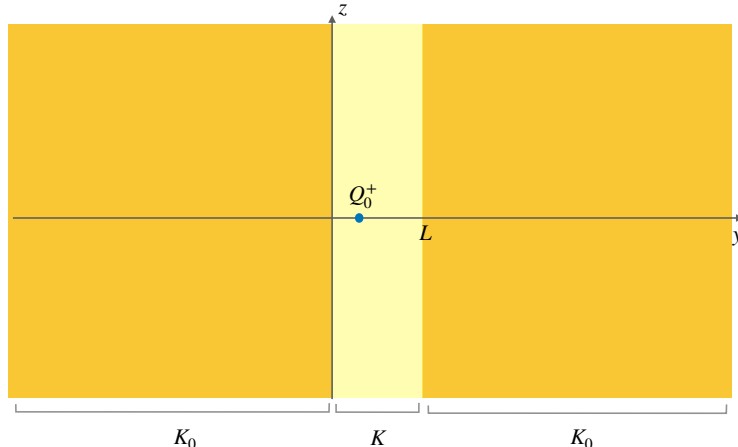

Figure 1: Space-time representation of a piecewise-homogeneous 2D gaussian free field (7) with alternating value of the coupling parameter, $K_0 - K - K_0$. Finding the propagator of the theory is equivalent to finding the potential generated by a unit charge $Q_0^+ = 1$ within a material characterized by dielectric constant $\frac{1}{\pi K}$ (yellow strip) surrounded by a material with different dielectric constant $\frac{1}{\pi K_0}$ (orange). In the quench problem (Sec. 4), $z = x$ is the spatial direction, while $y = \tau$ is the (imaginary) time direction. In the problem of a TLL coupled to leads (Sec. 6), instead, the role of the coordinates is reversed.

In this work we consider a particular situation, where $K(y,z)$ is chosen to be piecewise-homogeneous, as in Fig. 1, with a central strip characterised by a Luttinger parameter $K$ and two semi-infinite planes with parameter $K_0$. In this case a fully analytical solution can be found by exploiting the electrostatic analogy with the problem of a dielectric material with piece-wise homogeneous dielectric constant, and relying on the method of charge images.

Our motivation here is again the study of quantum quenches in TLLs. In fact, back in 2006, Refs. [5,6] provided a solution for the quantum quenches in general CFTs starting from a massive (finite correlation) state. The quench was studied using an imaginary time path integral approach and conformal transformations, together with techniques of boundary CFTs [26,27], which inherit methods of classical electromagnetism. These papers provided some very general results for the space-time decay of correlation functions and generated an enormous following literature (see Ref. [28] for a review).

Unfortunately, the method of Refs. [5,6] does not apply to quenches starting from an initial critical (massless) state. The latter, instead, has been solved, always back in 2006 [29], for the special case of a Luttinger model (with central charge $c = 1$), i.e., a quench in the Luttinger parameter $K_0 \to K$ (see also [30–37]), but relying on the operator formalism (via Bogolioubov transformations). Since then, however, nobody provided its characterization in terms of a path integral approach, and the question (already posed in Ref. [29]) on whether it is possible to extend the method of Refs. [5,6] to account for massless initial states, in TLLs or general CFTs, remained unanswered.

In this work we bridge this gap in the case of TLLs, by first arguing that the path integral formulation of massless quenches requires to deal with the action of an inhomogeneous 2D gaussian free field with piece-wise homogeneous Luttinger parameter. Once established such relation, we can use the electrostatic solution of the latter problem to recover all the correlation functions after the quench from a massless state within a path integral approach in imaginary time. In order to really put massive and massless quenches on the same ground, we further show that also for the massive quench can be solved directly via a very similar electrostatic

analogy, where only the value of the charge images change, while their positions remain the same.

An interesting outcome of this path integral perspective on massless quenches is their connection with the celebrated problem of a Tomonaga-Luttinger liquid with a TLL parameter $K$ coupled to two semi-infinite leads with TLL parameter $K_0$. For this problem it was shown [38–40] that contrarily to what was incorrectly claimed initially [41] the conductance is depending on the TLL parameter $K_0$ of the *leads* and not the one of the system. The two settings, of the quench and of the conductance with leads turn out to be simply related by an exchange of space and (imaginary) time directions, as it will be clear in the following.

The manuscript is organized as follows. In Section 2 we summarize the path integral approach to quenches in TLLs. In Section 3 we report the electrostatic solution of the piece-wise homogeneous gaussian free field (whose derivation is detailed in Appendix A). In Section 4 we apply such solutions to get results for massless quenches in TLLs. In Section 5 we report on a similar electrostatic solution of the massive quench. Section 6 is an independent section about the problem of a TLL coupled to leads. In Section 7 we discuss which are the problems to which how our method straightforwardly applies, and explicitly work out the example of the massless quench in a system of finite size with both periodic and open boundary conditions. We conclude in Section 8 with a summary, some further comments on simple generalizations of our results, and future perspectives.

## 2 Path integral approach to quenches in Tomonaga-Luttinger liquids

In this section we introduce the path integral approach in imaginary time to quenches in Tomonaga-Luttinger liquids. Namely, we consider the problem of a one-dimensional many-body quantum system initialized in the state $|\psi_0\rangle$ which is then let evolve with the Luttinger liquid hamiltonian

$$\hat{H}[\hat{\phi}, \hat{\Pi}] = \frac{u}{2\pi} \int dx \left[ \frac{1}{K} (\partial_x \hat{\phi})^2 + K(\pi \hat{\Pi})^2 \right], \tag{2}$$

with the fields $\hat{\phi}$ and $\hat{\Pi}$ satisfying the commutation relations $[\hat{\phi}(x), \hat{\Pi}(x')] = i\hbar \delta(x-x')$, and where $\{u, K\}$ are the sound velocity and Luttinger parameter, respectively, which fully define the model. Below we are going to set $\hbar = 1$ and $u = 1$.

We are interested in computing the expectation values of operators at some time $t$ after the quench. In particular, the expectation value of a local operator $\hat{O}(x)$ at time $t$ can be written as

$$\langle \hat{O}(x,t) \rangle = \lim_{\epsilon \to 0} \frac{\langle \psi_0 | e^{-\hat{H}\epsilon} e^{i\hat{H}t} \hat{O}(x) e^{-i\hat{H}t} e^{-\hat{H}\epsilon} | \psi_0 \rangle}{\langle \psi_0 | e^{-2\epsilon \hat{H}} | \psi_0 \rangle}, \tag{3}$$

where we introduced a damping factor $e^{-\epsilon \hat{H}}$, with $\epsilon > 0$, in such a way to make the path-integral expression convergent. In the path-integral formalism in imaginary time, the numerator in Eq. (3) can be represented as

$$\int [D\phi(x,\tau)] \langle \phi(x,\tau_1) | \psi_0 \rangle \langle \psi_0 | \phi(x,\tau_2) \rangle O(x,\tau=0) e^{-\int_{\tau_1}^{\tau_2} d\tau \mathcal{L}(\tau)}, \tag{4}$$

where $\int_{\tau_1}^{\tau_2} d\tau \mathcal{L}(\tau)$ is the euclidean action associated to the TLL, $|\phi(x,\tau)\rangle$ is the coherent states basis, and $\tau_{1,2}$ are supposed to be real and only at the end should be analytically continued to $\pm \epsilon - it$. At this point, Eq. (4) represents a path-integral over a strip of width $\tau_1 - \tau_2 = 2\epsilon$, with the operator $\hat{O}(x, \tau = 0)$ in path integral formalism (namely, $O(x, \tau = 0)$) inserted at $\tau = 0$,

while the initial state $|\psi_0\rangle$ plays the role of boundary condition. Depending on the nature of the initial state $|\psi_0\rangle$, however, such geometry gets modified as we are now going to explain.

## 2.1 Massive quench

If the initial state $|\psi_0\rangle$ is a massive state, namely has a finite correlation length, using renormalization group (RG) arguments [42], it can be replaced by the RG-boundary state $|B\rangle$ to which it flows. Effectively, this is taken into account at leading order by introducing an *extrapolation length* $\tau_0$, namely one makes the replacement $|\psi_0\rangle \propto e^{-\tau_0 \hat{H}}|B\rangle$, so that Eq. (3) becomes

$$\langle \hat{O}(x,t)\rangle \simeq \frac{\langle B|e^{-\hat{H}\tau_0}e^{i\hat{H}t}\hat{O}(x)e^{-i\hat{H}t}e^{-\hat{H}\tau_0}|B\rangle}{\langle B|e^{-2\hat{H}\tau_0}|B\rangle}, \tag{5}$$

where, since $\tau_0 > 0$, we could safely take the limit $\epsilon \to 0$. The equation above can be represented in a similar way in the $(x, \tau)$ plane (with $\tau$ being the imaginary time) as a path-integral over a strip, but this time of width $2\tau_0$ with boundary condition fixed by $|B\rangle$.

As pointed out in Refs. [5, 6], since both the theory (2) and the boundary state $|B\rangle$ are conformally invariant, one can use conformal maps and transformation of operators under those to evaluate the expectation value in the r.h.s. of Eq. (5) exactly. In particular this path-integral approach paved the way for the determination of several entanglement measures [43–45] that otherwise are very cumbersome to obtain even numerically in the operator approach [12, 46–48].

## 2.2 Massless quench

A different class of quenches is that starting from massless states, where, e.g., $|\psi_0\rangle$ is the ground state of the Tomonaga-Luttinger liquid hamiltonian $\hat{H}_0$ with $K_0 \neq K$ (cf. (2)). This means that in this case we are looking at a quench in the Luttinger parameter, i.e., $K_0 \to K$. While this problem has been solved in Ref. [29] by using Bogoliubov transformations, the explicit formulation and solution in the path-integral approach has not been yet worked out. This is what we do in the following.

The initial state can now be viewed as $|\psi_0\rangle \propto \lim_{\beta\to\infty} e^{-\beta \hat{H}_0}|\psi\rangle$ with some generic state $|\psi\rangle$ (which has a non-zero overlap with the ground state). Indeed, the action of the limit in $\beta$ is to project onto the ground state of $\hat{H}_0$. Eq. (3) therefore becomes

$$\langle \hat{O}(x,t)\rangle = \lim_{\beta\to\infty} \frac{\langle \psi|e^{-\hat{H}_0\beta}e^{-\hat{H}\epsilon}e^{i\hat{H}t}\hat{O}(x)e^{-i\hat{H}t}e^{-\hat{H}\epsilon}e^{-\hat{H}_0\beta}|\psi\rangle}{\langle \psi|e^{-2\hat{H}_0\beta}|\psi\rangle}, \tag{6}$$

which is the path-integral over a plane with an inhomogeneous Luttinger parameter (the imaginary evolution is in fact governed by two different hamiltonians, $\hat{H}$ and $\hat{H}_0$). Specifically, the Luttinger parameter is equal to $K$ in a slab of size $2\epsilon$ and $K_0$ otherwise. The geometry is the one shown in Fig. 1, with $(z, y) = (x, \tau)$, and $L = 2\epsilon$. The discontinuity is along the imaginary time, while along the spatial direction the Luttinger parameter is uniform.

At this point let us stress an important difference between the massive and massless quench. The action of the massive quench is defined within a strip of finite width with appropriate boundary conditions. This allows one to map (conformally) the geometry to the upper half plane and *then* use the analogy with electrostatic and the method of charge images to solve the problem. The geometry of the massless quench is quite different because it concerns a strip (whose length will be sent to zero at the end) within two semi-infinite planes. This means that an analogous transformation as the one used for the massive quench to map the strip into the upper half plane cannot be used. For this reason here we will approach the problem via its electrostatic analogy directly for the strip, without invoking any conformal mapping. For comparison, we will also show that the same strategy can be applied to the massive quench.

## 3  The general problem

In the previous section we reduced the problem of studying a massless quench to that of computing the path integral of an *inhomogeneous* free gaussian theory in two dimensions, whose Euclidean action reads

$$S = \frac{1}{2\pi} \int_\Omega dy\,dz \frac{1}{K(y,z)} \left[ (\partial_y \phi)^2 + (\partial_z \phi)^2 \right], \tag{7}$$

with $\Omega = \mathbb{R}^2$ defining the space where the theory lives. In our special case, $K(y,z)$ is a piecewise homogeneous function, namely it has the form (see Fig. 1)

$$K(y,z) = \begin{cases} K & 0 < y < L, \\ K_0 & \text{otherwise}, \end{cases} \tag{8}$$

so it is constant in the $z$-direction and has an alternating discontinuity in $y$.

Since the theory is quadratic, solving it amounts to finding the propagator $G(y,z;y',z') = \langle \phi(y,z)\phi(y',z') \rangle$. The latter satisfies the Poisson equation:

$$-\left[ \partial_y \frac{1}{K(y,z)} \partial_y + \partial_z \frac{1}{K(y,z)} \partial_z \right] G(y,z;y',z') = \pi \delta(y-y')\delta(z-z'), \tag{9}$$

and, in the electrostatic analogy, represents the electrostatic potential at position $(y,z)$ generated by a unit charge in $(y',z')$.

Because of the form (8) of $K(y,z)$, the propagator $G$ satisfies the Poisson equation (9) with constant $K$ in each domain and at the boundaries one has to impose the continuity of $G(y,z;y',z')$ and $\frac{1}{K(y,z)}\partial_y G(y,z;y',z')$ when $y \to 0^\pm$ and $y \to L^\pm$. These are the standard conditions that an electrostatic potential has to satisfy at the boundary between two materials with different dielectric constant $\frac{1}{\pi K_0}$ and $\frac{1}{\pi K}$ [49]. In the following, we will compute the propagator of this inhomogeneous system via the method of charge images. It is then worth recalling that the 2D electrostatic potential generated by a point charge $Q_P$ at $(y_P, z_P)$ in a uniform dielectric medium is

$$V_{2D}^P(y,z;y_P,z_P) = -Q_P \frac{K}{4} \log \frac{|(y-y_P)^2 + (z-z_P)^2|}{a^2}, \tag{10}$$

with $a$ a cutoff distance, which describes the equally well-known space dependence of the propagator in CFT. Note that while this propagator is infinite in $(y,z) = (y_P, z_P)$, in field theory this is usually regularized via a short-distance cutoff.

### 3.1  Solution

The method of charge images amounts to finding the solution of the original problem via the introduction of fictitious charges which guarantee the right boundary conditions at $y = 0, L$ and so, by uniqueness of the solution, generate the potential of interest [49]. While the solution for two half planes with different dielectric constants can be directly found in [49], for the problem of the strip one has to iterate the method by finding the correct set of charges that impose the desired boundary conditions at the edges of the strip and guess its asymptotic solution. In Appendix A we outline the first steps of such iteration, while below we only state the full solution. Likely this solution can be found in some textbooks, but it is easier to re-derive rather than searching for it.

We find that the potential inside the strip generated by a unit charge at position $(y',z')$ inside the strip (i.e., the propagator) can be interpreted as the potential of a uniform medium

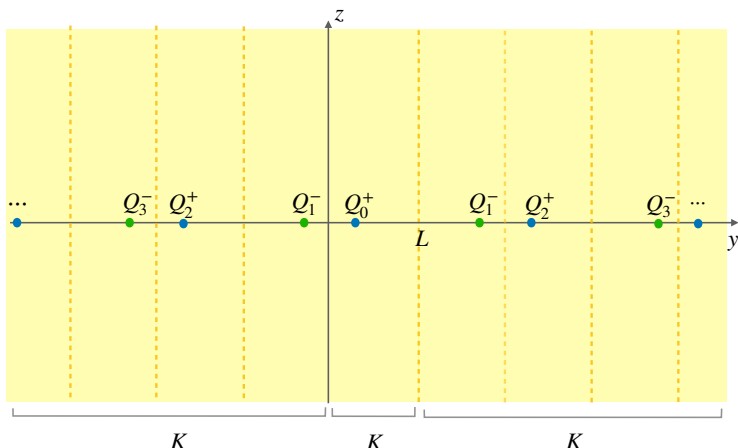

Figure 2: Position of the charges defined in Eqs. (11)-(12) that solve the piecewise-homogeneous gaussian field problem with alternating value of the coupling parameter, $K_0 - K - K_0$ (see Eqs. (7)-(9) in the main text).

of dielectric constant $\frac{1}{\pi K}$ generated by an infinite sequence of positive and negative charges with values and positions as follows

$$Q_{2|n|}^+ = \lambda^{2|n|} > 0 \qquad (y_n^+ = y' + 2Ln, z') \quad \text{for} \quad n \in \mathbb{Z}, \tag{11}$$

and

$$Q_{2|n|-1}^- = -\lambda^{2|n|-1} \qquad \begin{cases} (y_n^- = -y' + 2nL, z') & \text{for} \quad n > 0, \\ (y_n^- = -y' + 2(n+1)L, z') & \text{for} \quad n < 0, \end{cases} \tag{12}$$

with $\lambda = (K - K_0)/(K + K_0)$. Note that, except for $Q_0^+ (= 1)$, they all come in pairs.

In Fig. 2 we show the location of the charges. This implies that the overall potential can be found as the superposition of the logarithmic potential generated by each charge in a medium with dielectric constant $\frac{1}{\pi K}$. Note that, as expected, if $K = K_0$ there is only the charge in the slab with unit value.

The propagator (electrostatic potential) is thus given by

$$G(y, z; y', z') = \langle \phi(y, z)\phi(y', z') \rangle = -\frac{K}{4} \sum_{n \in \mathbb{Z}} Q_{2|n|}^+ \log \frac{|(z - z')^2 + (y - y_n^+)^2|}{a^2}$$

$$-\frac{K}{4} \sum_{n \in \mathbb{Z} \setminus \{0\}} Q_{2|n|-1}^- \log \frac{|(z - z')^2 + (y - y_n^-)^2|}{a^2} + C, \tag{13}$$

where $C$ is a constant. This is fixed by imposing some condition for $(y, z)$ on the boundary $\partial \Omega$ of the space $\Omega$ where the theory lives. In this case, as mentioned, $\Omega = \mathbb{R}^2$ and for $y, z \to \pm\infty$ the logarithm in (13) would diverge. Therefore we need to regularize $G(y, z; y', z')$ by considering an infinite additive constant ($|C| \to \infty$).

## 4 Results for the massless quench

In order to specialize the general solution of Sec. 3 to the massless quench problem, we take $z = x$, the spatial direction, and $y = \tau$, the imaginary time axis. As we mentioned, for this problem we work with a strip of size $L = 2\epsilon$ and eventually will take $\epsilon \to 0$. Note that if we

imagine to send $L \to 0$ all positive charges shrink on the same point, as well negative charges (cf. Eqs.(11)-(12)). This implies that if we place a unit positive charge in $(x, \tau)$ we will get two effective charges sitting at $(x, \tau)$ and $(x, -\tau)$ of charge

$$Q_{tot}^+ = (1 + 2\lambda^2 + 2\lambda^4 + \dots) = \left(2\sum_n \lambda^{2n} - 1\right) = \frac{1}{2}\left(\frac{K_0}{K} + \frac{K}{K_0}\right) = \mu_+, \qquad (14)$$

$$Q_{tot}^- = -2(\lambda + \lambda^3 + \lambda^5 + \dots) = -2\lambda\sum_n \lambda^{2n} = \frac{1}{2}\left(\frac{K_0}{K} - \frac{K}{K_0}\right) = -\mu_-. \qquad (15)$$

We want to compute $k$-points correlation functions of operators in the theory at time $t > 0$ after the quench. Those are derivatives, $\partial_x \hat{\phi}(x, t)$ and $\partial_t \hat{\phi}(x, t)$, and vertex operators $\hat{V}_\alpha(x, t) =: e^{i\alpha\hat{\phi}(x,t)}:$.

We start by considering the correlations functions of the derivative operators at equal time. We focus on $k = 2$, while the generalisation to $k > 2$ is straightforward. In this case, we define

$$J^{(2)}(x, t; 0, t) \equiv \langle \partial_x \hat{\phi}(x, t) \partial_{x'} \hat{\phi}(x', t) \rangle|_{x'=0}, \quad D^{(2)}(x, t; 0, t) \equiv \langle \partial_t \hat{\phi}(x, t) \partial_s \hat{\phi}(0, s) \rangle|_{s=t}. \qquad (16)$$

Moving to imaginary time, they can be readily written in terms of the propagator (9) specialised to the quench as

$$J^{(2)}(x, \tau; 0, \tau) = \partial_x \partial_{x'} G(\tau, x; \tau, x')|_{x'=0}, \quad D^{(2)}(x, \tau; 0, \tau) = -\partial_\tau \partial_\sigma G(\tau, x; \sigma, 0)|_{\sigma=\tau}. \qquad (17)$$

In the limit $\epsilon \to 0$, and after considering the analytic continuation $\tau \to it$, we get the following equal-time correlations

$$J^{(2)}(x, t; 0, t) = -\frac{K\mu^+}{2}\frac{1}{x^2} + \frac{K\mu^-}{4}\left[\frac{1}{(x-2t)^2} + \frac{1}{(x+2t)^2}\right], \qquad (18)$$

$$D^{(2)}(x, t; 0, t) = -\frac{K\mu^+}{2}\frac{1}{x^2} - \frac{K\mu^-}{4}\left[\frac{1}{(x-2t)^2} + \frac{1}{(x+2t)^2}\right], \qquad (19)$$

which show a power law behavior, as expected from the massless nature of the initial state.

We then move to the correlation functions of vertex operators

$$C_{\{\alpha_j\}}^{(k)} \equiv \langle \hat{V}_{\alpha_1}(x_1, \tau) \cdots \hat{V}_{\alpha_k}(x_k, \tau) \rangle. \qquad (20)$$

The easiest way to compute it is to recall that, in the electrostatic analogy, the logarithm of (20) is (up to a sign) the electrostatic potential energy of a system of point charges $\{\alpha_1, \cdots, \alpha_k\}$ [11], i.e., $\log C_{\{\alpha_j\}}^{(k)} = -U_{\{\alpha_j\}}^{(k)}$. Therefore, we need to modify the source term (r.h.s.) of the Poisson equation (9), that now takes the form of a sum of delta functions at the positions of the charges. By linearity, the solution of such modified equation, namely the total potential $G(y, z)$, will be given by

$$G(y, z) = \sum_{i=1}^k \alpha_i G(y, z; y_i, z_i). \qquad (21)$$

Now, the total electrostatic potential energy of the system can be written as

$$U_{\{\alpha_j\}}^{(k)} = \frac{1}{2}\int_\Omega \mathrm{d}y\mathrm{d}z\, E(y, z) \cdot D(y, z) - \frac{1}{2}\sum_{i=1}^k \alpha_i \lim_{(y,z)\to(y_i,z_i)} V_{2D}^i(y, z; y_i, z_i), \qquad (22)$$

where we defined the electric field $E = -\nabla G(y, z)$, and the displacement field $D(y, z) = \frac{1}{\pi K(y,z)} E(y, z)$ [49]. The last term (cf. Eq. (10)) cancels the self-interaction which

corrects the first term and it is needed because in our case we have a set of point-like charges. The integration can be carried out by parts

$$\int_\Omega \mathrm{d}y\mathrm{d}z\, E(y,z)\cdot D(y,z) = \sum_{i=1}^{k} \alpha_i G(y_i,z_i)\,, \tag{23}$$

where we used that $G(y,z) = 0$ on the boundary $\partial\Omega$, and we evaluated $\nabla\cdot D$ via the Poisson equation. Therefore

$$U_{\{\alpha_j\}}^{(k)} = \frac{1}{2}\sum_{i=1}^{k}\alpha_i \lim_{(y,z)\to(y_i,z_i)}\left[G(y,z)-V_{2D}^i(y,z;y_i,z_i)\right] \equiv \frac{1}{2}\sum_{i=1}^{k}\alpha_i G_i(y_i,z_i)\,, \tag{24}$$

where we defined the function $G_i(y_i,z_i)$ as the potential generated by all charges (real and imaginary) that here we denote as the set $\{\alpha_j Q_m^s\}$, with $s = \pm$, except the charge $\alpha_i Q_0^+ = \alpha_i$ itself, i.e.,

$$G_i(y,z) = -\frac{K}{2}\left(\sum_{(j,m,s)\backslash(i,0,+)}\alpha_j Q_m^s \log\frac{|r_{i,0}^+ - r_{j,m}^s|}{a} + C\right)\,, \tag{25}$$

where $r_{j,m}^s = (y_{j,m}^s, z_{j,m}^s)$ are the positions of the image charges associated to $\alpha_j$ (in particular $r_{j,0}^+$ is the position of $\alpha_j$ itself).

In order to write an explicit expression, we now specify to $k = 2$, namely the two point function

$$C_\alpha^{(2)}(x,\tau;0,\tau) \equiv \langle V_\alpha(x,\tau)V_{-\alpha}(0,\tau)\rangle\,. \tag{26}$$

So, in the electrostatic problem, we have two charges with value $\pm\alpha$ at position $(x,\tau)$ and $(0,\tau)$ inside the strip. The potential energy is given by (24) together with (25) with $k = 2$ and $\alpha_1 = -\alpha_2 = \alpha$. As mentioned, however, in the $\epsilon\to 0$ limit, many image charges shrink on the same point. Eventually, the potential energy can be effectively computed as the sum of the following contributions:

- the one of the charge $\alpha$ in $(x,\tau)$ due to the potential generated by the charge $-\alpha\mu^+$ in $(0,\tau)$, the charge $\alpha\mu^-$ in $(x,-\tau)$, and $-\alpha\mu^-$ in $(0,-\tau)$;

- the one of the charge $\alpha$ in the strip and its charge images with the same sign $\alpha Q_{2|n|}^+$ (the latter, however, is associated to a distance $\log|4\epsilon n|$ divergent when $\epsilon\to 0$ and independent on $x$ and $\tau$);

- the same contributions for the potential energy of the charge $-\alpha$.

By summing all such contributions one gets (at leading order in $\epsilon$)

$$U_\alpha^{(2)}(x,\tau;0,\tau) = \frac{1}{2}\alpha^2\mu^+ K\log\frac{|x+a|}{a} - \frac{1}{2}\alpha^2\mu^- K\log\frac{|x+i2\tau|}{a} + \frac{1}{2}\alpha^2\mu^- K\log\frac{|2\tau|}{a}$$
$$+ \frac{1}{2}\alpha^2(\mu^+ - 1)\log\frac{\epsilon}{a} + \mathrm{const}\,, \tag{27}$$

where we explicitly introduced the short-distance cutoff $a$. Note that the (infinite) constant $C$ in the potential cancels because the total (real) charge is zero. Note also that the same would not true for the one-point function, which due to the presence of such infinite constant is instead zero (as it should).

The result for the correlation function (26) is finally given by

$$C_\alpha^{(2)}(x,\tau;0,\tau) = e^{-U_\alpha^{(2)}(x,\tau;0,\tau)}\,, \tag{28}$$

where in the last expression the dependence on $\epsilon$ has disappeared. Now, to get its behavior in real time, the last step is to consider the analytic continuation $\tau \to it$, leading to the equal-time correlation after the quench

$$C_\alpha^{(2)}(x,t;0,t) = \left( \left| \frac{x}{a} \right|^{-\mu^+ K/2} \left| 1 - \left( \frac{x}{2t} \right)^2 \right|^{\mu^- K/4} \right)^{\alpha^2}, \qquad (29)$$

where we only kept the leading order in $a$. So, again, we find that it decays as a power-law. We stress that our results perfectly match those obtained in Refs. [29, 50–52] by using the real-time operator approach.

## 5 Electrostatic solution of the massive quench: a comparison

In this section we show how to solve the massive quench without invoking any conformal transformation, via its analogy with an electrostatic problem similar to the one used for the massless quench: this is very useful, because it really puts the two types of quenches on equal grounds.

The main differences are that now the size of the strip is finite $2\tau_0$ and, most importantly, the boundary conditions that one has to impose at the edges of the strip are different. In fact for the massive quench one has to solve the *homogeneous* Poisson equation

$$\nabla^2 G(y,z;y',z') = -\pi K \delta(y - y') \delta(z - z'), \qquad (30)$$

within the strip domain $\Omega$, and impose at the boundary $\partial\Omega$ (now given by the boundaries of such strip) the appropriate boundary conditions compatible with the chosen boundary CFT. In the following we focus on the boundary condition $G(y = 0, z) = G(y = 2\tau_0, z) = 0$, which is one of the possible conformal invariant boundary conditions [26]. The other possible choices require minor adjustments that are easily taken into account and not worth discussing.

Thus, for $G = 0$ at the boundaries, we end up with the electrostatic problem of a dielectric between two conducting lines (see Fig. 3 (a)). This problem can be solved, again, via the method of charge images, and the solution is given by a sequence of charges, as shown in Fig. 3 (b) [49]. The position of these charges is the same as those in Fig. 2, the only difference being that now their value is the same for all positive and negative charges. Moreover, the overall constant in the propagator here is set to zero in order to satisfy the boundary conditions. at the edges of the strip.

For the quench problem, again, in Fig. 3 one sets $y = \tau$, $z = x$ and now $L = 2\tau_0$. With the same logic of the previous section, one can now use this electrostatic picture to compute correlation functions. Below we focus in particular on the vertex operators.

In order to compute the one point function

$$C_\alpha^{(1)}(0, \tau) = \langle e^{i\alpha\hat{\phi}(0,\tau)} \rangle, \qquad (31)$$

we consider the potential energy of a charge in the strip at position $(0, \tau)$ due to all other (image) charges. Applying Eq. (22) we obtain

$$U_\alpha^{(1)}(0, \tau) = -\frac{\alpha^2 K}{4} \left[ \sum_{n \neq 0} \log \frac{|4\tau_0 n|}{a} - \sum_n \log \frac{|2\tau + 4\tau_0 n|}{a} \right], \qquad (32)$$

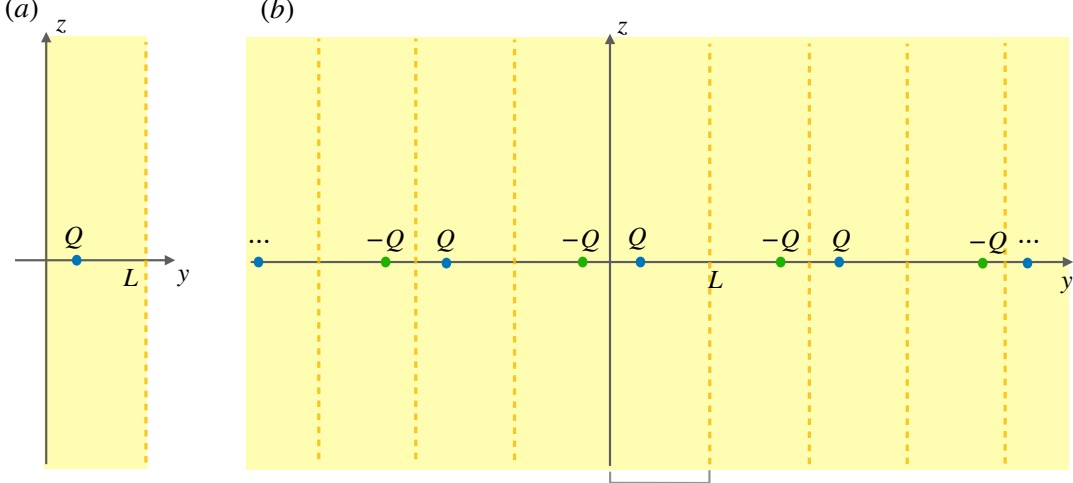

Figure 3: Panel (a) : Problem of a charge $Q$ in a strip of dielectric material between two conducting lines at $y = 0, L$. Panel (b) : Position and values of the image charges that give the correct electrostatic potential inside the strip. This is the electrostatic analog of a massive quench problem, where the propagator plays the role of the electrostatic potential (see discussion in the text). To be compared with Fig. 2.

and therefore

$$C_\alpha^{(1)}(0,\tau) = e^{-U_\alpha^{(1)}(0,\tau)} = \left[ \frac{a}{2(\tau + 2\tau_0 n)}\bigg|_{n=0} \prod_{n>0} \frac{4\tau_0^2 n^2}{|\tau^2 - 4\tau_0^2 n^2|} \right]^{\frac{K\alpha^2}{4}} = \left[ \frac{\pi a}{4\tau_0} \frac{1}{\sin(\frac{\pi\tau}{2\tau_0})} \right]^{\frac{K\alpha^2}{4}}. \tag{33}$$

We thus recognize the one point function obtained in [6] via conformal maps, which, upon analytic continuation gives an exponential decay in time.

In order to compute the two point function at equal time

$$C_\alpha^{(2)}(x,\tau;0,\tau) = \langle e^{i\alpha\hat\phi(x,\tau)} e^{-i\alpha\hat\phi(0,\tau)} \rangle \tag{34}$$

we place a positive charge $\alpha > 0$ in $(x,\tau)$ and a negative one $-\alpha$ at $(0,\tau)$. There will be then two sets of image charges. We thus compute the potential energy of these two charges in the strip, given by (again, we reintroduce the short-distance cutoff $a$)

$$\begin{aligned} U_\alpha^{(2)}(x,\tau;0,\tau) = &\frac{1}{2}\alpha^2 K \sum_n \log \frac{|x + a + i(4\tau_0 n)|}{a} \\ &- \frac{1}{2}\alpha^2 K \sum_n \log \frac{|x + a + i(2\tau + 4\tau_0 n)|}{a} + 2U_\alpha^{(1)}(0,\tau). \end{aligned} \tag{35}$$

This gives the two point function as

$$\begin{aligned} e^{-U_\alpha^{(2)}(x,\tau;0,\tau)} &= \left[ \frac{|(x+a) + i2\tau|^2}{(x+a)^2} \bigg| \prod_{n>0} \frac{(x+a+i2\tau)^2 + 16\tau_0^2 n^2}{(x+a)^2 + 16\tau_0^2 n^2} \bigg|^2 \right]^{\frac{\alpha^2 K}{4}} e^{-2U_\alpha^{(1)}(0,\tau)} \\ &= \left( \frac{|\sinh \frac{\pi(x+a+i2\tau)}{4\tau_0}|^2}{|\sinh \frac{\pi(x+a)}{4\tau_0}|^2} \right)^{\frac{K\alpha^2}{4}} e^{-2U_\alpha^1(0,\tau)} \simeq \left[ \left(\frac{\pi a}{4\tau_0}\right)^2 \frac{\cosh \frac{\pi x}{2\tau_0} - \cos \frac{\pi\tau}{\tau_0}}{2\sinh^2 \frac{\pi(x+a)}{4\tau_0} \sin^2 \frac{\pi\tau}{2\tau_0}} \right]^{\frac{K\alpha^2}{4}}, \end{aligned} \tag{36}$$

where by $\simeq$ we mean the leading order in $a$.

Again, we recover the result obtained in [6], and the analytic continuation allows us to recover the real time behavior.

# 6 An apparently unrelated problem: Tomonaga-Luttinger liquid coupled to leads

Quite remarkably, the problem of the massless quench that we have studied in Sec. 4, turns out to be intimately related to the apparently different problem considered in the series of works [38–40]. These works deal with the study a one dimensional conductor, a Tomonaga-Luttinger liquid with parameter $K$, coupled to two semi-infinite leads, modelled as semi-infinite TLLs with parameter $K_0$. The main focus of these papers is the study of the dc conductivity of the system, within linear response.

In a path integral formulation one can immediately see the similarity between the two problems: in fact this one corresponds to consider a discontinuity of the parameter $K$ along the $x$ axes while keeping it constant along $\tau$. The solution of the propagator is therefore the same with the exchange of space and time directions.

One can therefore use the general result of Sec. 3 for the imaginary-time propagator $G$ with $y = x$ and $z = \tau$ to derive the result of [38–40] for the conductivity, following a similar path to that described in [39].

In order to get the conductivity $\sigma_\omega(x, x')$, one just needs to Fourier transform the imaginary time propagator $G(x, \tau; x', 0)$ in frequency and take the $\omega \to 0$ limit of that propagator. In fact it holds [38–40]

$$\sigma_\omega(x, x') = -e^2 \frac{\overline{\omega}}{\pi} G_{\overline{\omega}}(x, x'),$$ (37)

with $\omega = i\overline{\omega} + \epsilon$.

To get the Fourier transform, we use that

$$\int_{-\infty}^{\infty} e^{-i\omega\tau} \log[a^2 + \tau^2] = -\frac{2\pi}{|\omega|} e^{-a|\omega|} - 4\pi\gamma_E \delta(\omega),$$ (38)

obtaining

$$\mathcal{F}_{\tau\to\omega}[G(x, \tau; x', 0)] = \frac{K}{2} \frac{\pi}{|\omega|} \left\{ \sum_{n\in\mathbb{Z}} \left(\frac{K - K_0}{K + K_0}\right)^{2|n|} e^{-|\omega(x - x' - 4nL)|} \right.$$
$$\left. - \sum_{n\in\mathbb{Z}\setminus\{0\}} \left(\frac{K - K_0}{K + K_0}\right)^{2|n|-1} e^{-|\omega(x + x' - 4nL)|} \right\}.$$ (39)

This reproduces the ansatz given, e.g., in Ref. [38] for the propagator. Let us just mention that as we give all the details of the propagator one could read it at *arbitrary* frequency. However, for the purpose of computing the dc conductivity one is interested in the $\omega \to 0$ limit only [39]. This reads

$$\lim_{\omega\to 0} \frac{\omega}{\pi} \mathcal{F}_{\tau\to\omega}[\langle \phi(x, \tau)\phi(x', 0)\rangle] = \frac{K}{2} \left[ \sum_{n\in\mathbb{Z}} \left(\frac{K - K_0}{K + K_0}\right)^{2|n|} - \sum_{n\in\mathbb{Z}\setminus\{0\}} \left(\frac{K - K_0}{K + K_0}\right)^{2|n|-1} \right]$$ (40)
$$= \frac{K}{2}[\mu^+ - \mu^-] = \frac{K_0}{2}.$$

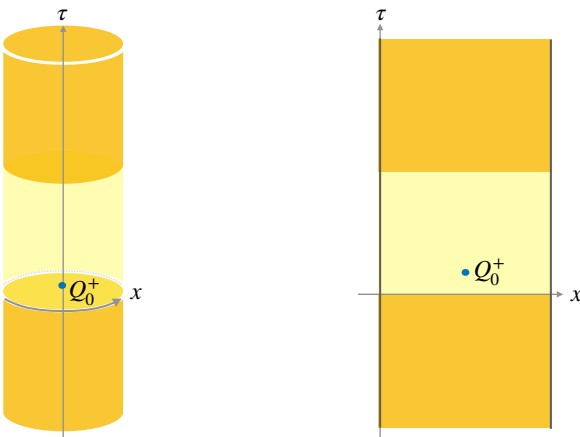

Figure 4: Path integral geometries corresponding to the massless quench in finite system size: PBC give rise to the cylinder geometry on the left, while OBC to the strip geometry on the right.

This result shows that the conductivity only depends on the properties of the leads, in agreement with the results of [38–40].

Our method allows for generalization to either frequency dependence or to more complicated cases. These will be discussed elsewhere.

# 7 Massless quench in finite systems

It is straightforward to generalise our results to study problems that within the path integral formulation admit the following structure: the associated worldsheet only has two straight interfaces in one of the two directions. In this section, in particular, we take this direction to be the imaginary time, and focus on the massless quench in system of finite size $L$. This amounts to consider the space direction to be finite, ending up in a cylinder (PBC) or in a strip (OBC) geometry, as showed in Fig. 4. In both cases, all the results discussed in Section 4 can be derived exactly as in the case of infinite system size, while only the core building block, namely the homogeneous propagator (10) has to be replaced by the corresponding one in finite system size (with the appropriate boundary conditions). Below we report directly some final useful formulas.

## 7.1 Periodic boundary conditions

When considering a finite system with PBC the only change in the computations in the previous section is to replace the homogeneous propagator (10) of an infinite system with its counterpart in the cylinder, i.e.

$$V^Q_{\mathrm{PBC}}(x_1, \tau_1; x_2, \tau_2) = -\frac{KQ}{4} \log\left(\sin^2\left[\frac{\pi}{L}(x_1 - x_2)\right] + \sinh^2\left[\frac{\pi}{L}(\tau_1 - \tau_2)\right]\right). \quad (41)$$

From the above equation, we can simply get the full propagator after a massless quench $K_0 \to K$ by considering the geometry in Fig. 4 (left). The interfaces along the $\tau$-axis will give rise to the usual charge images $\{Q^s_m\}$ at positions $\{\tau^s_m\}$ (with $s = \pm$), as given in Eqs. (11)-(12), and thus

$$G_{\mathrm{PBC}}(x, \tau; x', \tau') = \sum_m \sum_{s=\pm} V^{Q^s_m}_{\mathrm{PBC}}(x, \tau; x', \tau^s_m). \quad (42)$$

Finally the correlation functions after the quench are obtained upon analytic continuation $\tau \to it$. For example, the two-point correlation function of vertex operators now becomes

$$C_\alpha^{(2)}(x,t;0,t) = \left[ \left( \frac{1}{\sin^2 \frac{\pi x}{L}} \right)^{\mu^+} \left( 1 - \frac{\sin^2 \frac{\pi x}{L}}{\sin^2 \frac{2\pi t}{L}} \right)^{\mu^-} \right]^{K\alpha^2/4}, \tag{43}$$

again in agreement with the result in [29].

Finally, note that, by exchanging the role of space and time directions, the same solution allows to treat the problem of a conductor coupled to leads at finite temperature (and eventually get the ac conductivity) [53,54].

## 7.2 Open boundary conditions

One can also work out the solution for a massless quench in a finite system of size $L$ with OBC imposed at the boundaries, see Fig. 4 (right). More specifically, we choose Dirichelet boundary conditions, namely a vanishing bosonic field at $x = 0, L$.

Similarly to the PBC case, the full propagator after the quench can be written in terms of the one in the homogeneous strip, $V_{\text{OBC}}^Q$, as

$$G_{\text{OBC}}(x,\tau;x',\tau') = \sum_m \sum_{s=\pm} V_{\text{OBC}}^{Q_m^s}(x,\tau;x',\tau_m^s). \tag{44}$$

However, we further note that, by exchanging role of space and time coordinates, the propagator in the homogeneous strip was computed in Section 5 for the massive quench. The solution was given in terms of infinitely many charge images with alternating sign and usual positions $\{x_n^\pm\}$ with $x_n^\pm = \pm x' + 2Ln$ (with $n \in \mathbb{Z}$) (cf. Fig.3). In formulas

$$V_{\text{OBC}}^Q(x,\tau;x',\tau') = \sum_n \sum_{s'=\pm} V^{s'Q}(x,\tau;x_n^{s'},\tau), \tag{45}$$

where we used the following notation $V^Q(y,z;y_P,z_P) \equiv V_{2D}^P(y,z;y_P,z_P)$ (cf. (10)).

An interesting perspective comes out if we combine (44) and (45): the propagator $G_{\text{OBC}}$, written as two infinite sums (one in the space and one in the time directions), can be interpreted via a unique distribution of charge images as shown in Fig. 5.

Note that (45) can be resummed giving

$$V_{\text{OBC}}^Q(x,\tau;x',\tau') = -\frac{K}{4}Q\log\left( \frac{\cos\left[\frac{\pi(x-x')}{L}\right] - \cosh\left[\frac{\pi(\tau-\tau')}{L}\right]}{\cos\left[\frac{\pi(x+x')}{L}\right] - \cosh\left[\frac{\pi(\tau-\tau')}{L}\right]} \right). \tag{46}$$

Remarkably, in the case of the quench only (when considering the limit $\epsilon \to 0$) also the full propagator $G_{\text{OBC}}$ in (44) can be resummed.

The final step to get correlation functions is again to make the analytic continuation $\tau \to it$. We report as an example the result for two-point correlation function of vertex operators, which can be written as

$$C_\alpha^{(2)}(x,t;x',t) = \left[ \left( \frac{\cos\left[\frac{\pi(x+x')}{L}\right] - 1}{\cos\left[\frac{\pi(x-x')}{L}\right] - 1} \right)^{\mu^+} \left( \frac{\cos\left[\frac{\pi(x-x')}{L}\right] - \cos\left[\frac{2\pi t}{L}\right]}{\cos\left[\frac{\pi(x+x')}{L}\right] - \cos\left[\frac{2\pi t}{L}\right]} \right)^{\mu^-} \right.$$

$$\left. \times \left( \frac{1 - \cos\left[\frac{2\pi t}{L}\right]}{\cos\left[\frac{2\pi x}{L}\right] - \cos\left[\frac{2\pi t}{L}\right]} \right)^{-\mu^-/2} \left( \frac{1 - \cos\left[\frac{2\pi t}{L}\right]}{\cos\left[\frac{2\pi x'}{L}\right] - \cos\left[\frac{2\pi t}{L}\right]} \right)^{-\mu^-/2} \right]^{\frac{K\alpha^2}{4}}. \tag{47}$$

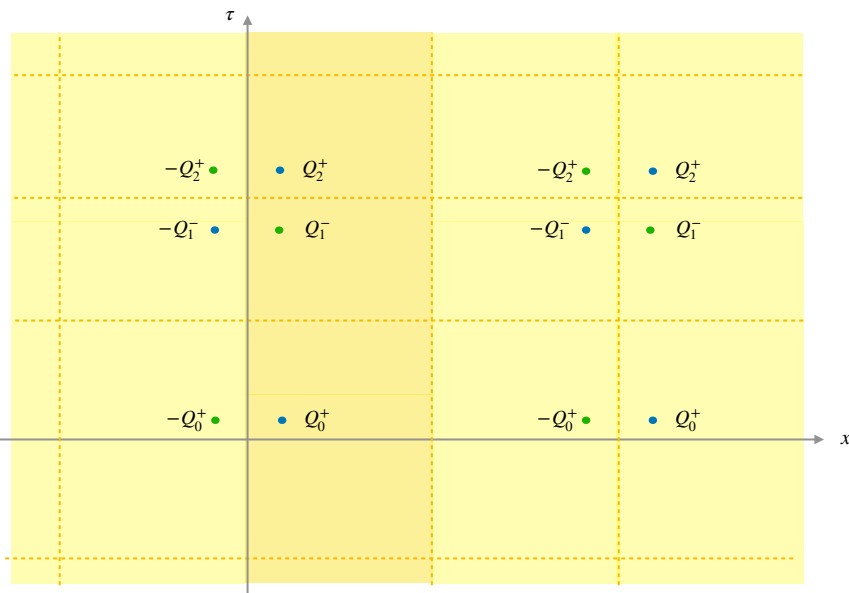

Figure 5: Distribution of charge images in the massless quench in a finite system with OBC. The physical strip corresponding to the one in Fig. 4 (right) is highlighted (light orange) for readability.

## 8 Conclusions

In this work we have reconsidered the problem of the (massless) quench of the Luttinger parameter in a Tomonaga-Luttinger model. This problem was already solved long ago in real-time by Bogolioubov transformations [29], but we present a new solution that works in the path integral formalism in imaginary time. This path integral formulation maps the quench into the electrostatic problem of a strip of a dielectric medium surrounded by another infinite dielectric. Our approach bridges a theoretical gap that allows one to understand on the same ground massive and massless quenches in Luttinger models. As a byproduct, we have shown that the quench problem is equivalent, after an exchange of space and time, to the one of a finite one dimensional conductor coupled to two semi-infinite leads. Our solution is very general and paves the way to the study of interfaces in other dimensions or geometries.

A part from the generalization to finite system quench considered in Sec. 7 (or the equivalent problems in the LL coupled to leads), another straifghtforward generalization concerns the $d$-dimensional problem. In fact, the values and positions of charge images (11)-(12) do not depend on the specific form of the potential generated by a single charge (this can be easily understood from the derivation in Appendix A). Thus, we can consider the problem of a heterogenous Gaussian theory in $d$ dimensions with $d-1$ dimensional interfaces and use the same solution changing only the specific form of the potential generated by each charge.

A less straightforward and more interesting outlook concerns the study of the entanglement (both in and out of equilibrium) in the presence of the permeable interface [55]. Such problem has been considered a lot in the literature [56–65], but we believe that the electrostatic analogy could lead to a simpler and more transparent solution.

## Acknowledgements

We thank Jérôme Dubail for useful discussions. This work is supported by "Investissements d'Avenir" LabEx PALM (ANR-10-LABX-0039-PALM) (EquiDystant project, L. Foini). PC acknowledges support from ERC under Consolidator grant number 771536 (NEMO). This work was supported in part by the Swiss National Foundation under division II.

## A Solution of the equivalent electrostatic problem

In this Appendix we report the solution of the electrostatic problem discussed in Sec. 3, namely the one of a strip of dielectric material, surrounded by a different material. This geometrical setting leads to a discontinuity in the dielectric constant, that as in Fig. 1, we assume to be along the $y$ direction. The value of the dielectric constant is $\epsilon = \frac{1}{\pi K}$ inside the strip of width $L$, and $\epsilon' = \frac{1}{\pi K_0}$ outside.

The solution is derived via the method of charge images which amounts to find the true potential introducing fictitious charges and considering the medium as homogenous. Such charge configuration will be find iteratively, and is inspired by the solution of a dielectric between two conducting lines (this is briefly reviewed in Section 5 in the context of its relation to massive quenches).

To fix the ideas, we suppose to have a charge $q$ at the point $(a, 0)$ with $0 < a < L$. The problem is translational invariant along $z$ so without loss of generality we set $z = 0$. We will enforce the continuity of the potential $V(y, z; a, 0)$ and of $\epsilon(y, z)\partial_y V(y, z; a, 0)$ across the interface. The goal is to find the potential $V_{in}(y, z; a, 0)$ within the strip, corresponding to the propagator $G(y, z; a, 0)$ in Eq. (9).

The potential generated by a charge $q$ in two dimensions is logarithmic. However, to show the generality of this approach we will consider a potential of the form $V(y, z; y', z') = \frac{1}{\epsilon} q f((y - y')^2 + (z - z')^2)$ where $f$ is a generic (smooth) function, and $(y', z')$ is the position of the charge.

*First step:* Let us first consider the interface at $y = 0$, and solve the problem while ignoring the presence of the second interface. In this case, the potential inside the strip is found by considering a single image charge, placed symmetrically to $q$ with respect to $y = 0$. Following [49, 66], this can be understood by considering in $0 \le y \le L$ a potential generated by two charges $q, q_1$ (the true one plus the image) in $a$ and $-b < 0$

$$V_{in}(y, z; a, 0) = \frac{q}{\epsilon} f(z^2 + (y - a)^2) + \frac{q_1}{\epsilon} f(z^2 + (y + b)^2), \tag{48}$$

and the potential outside $y < 0$ as generated by the charges $q, p_1$ as

$$V_{out}^-(y, z; a, 0) = \frac{q}{\epsilon} f(z^2 + (y - a)^2) + \frac{p_1}{\epsilon} f(z^2 + (y - c)^2). \tag{49}$$

The continuity of $V$ at $y = 0$ implies

$$b = c \qquad \text{and} \qquad q_1 = p_1. \tag{50}$$

The continuity of $\epsilon \, \partial_y V(y, z)$ implies

$$\left(q a f'(z^2 + (a)^2) - q_1 b f'(z^2 + (b)^2)\right) = \frac{\epsilon'}{\epsilon}\left(q a f'(z^2 + (a)^2) + q_1 b f'(z^2 + (b)^2)\right). \tag{51}$$

This gives

$$b = a \qquad \text{and} \qquad q_1 = -q\frac{\epsilon' - \epsilon}{\epsilon + \epsilon'}, \tag{52}$$

as anticipated.

*Second step:* We now focus on the other interface at $y = L$, and note that the solution (48) does not satisfy the continuity conditions at $y = L$. Therefore $V_{in}$ has to be modified. To do that, the idea is to "balance" the two charges $q, q_1$ generating the potential in the strip at the previous step, by placing two more charges symmetric to those with respect to the axis $y = L$. To see that, we consider inside the strip ($0 < y < L$) a potential of the form

$$V_{in}(y, z; a, 0) = \frac{q}{\epsilon} f(z^2 + (y-a)^2) + \frac{q_1}{\epsilon} f(z^2 + (y+a)^2) + \frac{q_2}{\epsilon} f(z^2 + (y-b)^2) + \frac{q_3}{\epsilon} f(z^2 + (y-d)^2),$$
(53)

with $b, d > L$, while for $y > L$ we take the following ansatz

$$V_{out}^+(y, z; a, 0) = \frac{q}{\epsilon} f(z^2 + (y-a)^2) + \frac{q_1}{\epsilon} f(z^2 + (y+a)^2) + \frac{p_2}{\epsilon} f(z^2 + (y-c)^2) + \frac{p_3}{\epsilon} f(z^2 + (y-e)^2),$$
(54)

with $c, e < L$. The continuity of $V$ in $y = L$ implies

$$c = 2L - b \qquad e = 2L - d \quad q_2 = p_2 \qquad q_3 = p_3.$$
(55)

The continuity of $\epsilon \partial_y V$ at $y = L$ implies:

$$\begin{aligned}
&\frac{\epsilon - \epsilon'}{\epsilon} \left( q(L-a) f'(z^2 + (L-a)^2) + q_1(L+a) f'(z^2 + (L+a)^2) \right) \\
&= \frac{\epsilon + \epsilon'}{\epsilon} \left( q_3(d-L) f'(z^2 + (L-d)^2) + q_2(b-L) f'(z^2 + (L-b)^2) \right),
\end{aligned}$$
(56)

namely

$$d = 2L - a \quad q_3 = q_1 = -q \frac{\epsilon' - \epsilon}{\epsilon + \epsilon'} \quad b = 2L + a \quad q_2 = -q_1 \frac{\epsilon' - \epsilon}{\epsilon + \epsilon'} = q \left( \frac{\epsilon' - \epsilon}{\epsilon + \epsilon'} \right)^2.$$
(57)

*Following steps:* Then one has to proceed iteratively. It is easy to realize that at each step a new pair of image charges has to be added on one of the two sides of the strip, alternatively, in order to satisfy the continuity conditions at $y = 0$ and $y = L$, respectively. While the positions of such pairs can be easily guessed by symmetry, the continuity conditions also fix the values of the image charges, which after a few steps can be guessed as well (and proved by induction). In this way, we finally arrive to the solution presented in Sec. 3 in Eq. (11)-(12). For comparison, one has to identify

$$Q_0^+ = q, \quad Q_1^- = q_1 = q_3, \quad Q_2^+ = q_2 = q_4, \quad \cdots$$
(58)

and set $q = 1$ (a unit charge).

Finally we note that our approach generalises even further. In fact one can take a function $f$ of the form

$$V(y, z; y', z') = \frac{1}{\epsilon} q f(y - y', z - z', z + z'),$$
(59)

even with respect to its arguments. This allows one to relax the constraint of translational invariance along the $z$ direction, as for the quench with open boundary conditions (cf. Sec. 7).

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
