# Peer review of "Electrostatic solution of massless quenches in Luttinger liquids"

_SciPost Physics, doi:SciPost Phys. 13, 111 (2022)_

## Round 2 · Referee Report · Anonymous (Referee 1) · 2022-4-17

Strengths

1. Clearly stated method and solution.
2. Undoubtedly correct as results agree with those from other methods.
3. Fills a gap in the literature.

Weaknesses

A rather narrow and pedestrian approach to the problem, which overlooks possibilities of generalizations to new results.

Report

This paper applies the methods of Refs (5,6) to the problem of the quench of the Luttinger parameter, taken as being described by a free massless boson.
Not surprisingly it reproduces the known results, and, as such, the authors might have made it more ... by deriving new ones, such as the case of boundaries or a completely finite system, which would be an interesting exercise in elliptic functions; or by comparing against other methods which work for free fields, eg using the method of stationary phase in real time, thereby giving a geometric optics quasiparticle interpretation to the results.

As it stands, this paper is not up to the high standards of SciPost.

Requested changes

1. Add more material with some new results using the same approach.

  • validity: high
  • significance: low
  • originality: low
  • clarity: high
  • formatting: excellent
  • grammar: excellent

Author:  Paola Ruggiero  on 2022-05-17  [id 2482]

(in reply to Report 1 on 2022-04-17)

We already replied to the main criticism about the novelty of our results (also appreciated in the second report) in our previous comment and we do not further discuss it here.
However, to meet the referee suggestion to include further results in our manuscript, we also added an entire section about the massless quench for finite systems both with periodic or open boundary conditions.
Although the goal of our work was to describe in detail a new method and establish connections between different problems, this new section provides an example of how this method generalizes to other settings.

Author:  Paola Ruggiero  on 2022-04-20  [id 2398]

(in reply to Report 1 on 2022-04-17)

We thank the referee for the report, though we are surprised by some of his/her comments and we disagree with his/her conclusions.

Let us start by stressing again that the main goal of our work is exactly to fill a rather long-standing gap in the literature, namely the imaginary-time path integral formulation of massless quenches in Luttinger Liquids, which is correctly acknowledged in the report as one of the strength of our paper. The interest in such problem was already raised in Ref. [29], back in 2006 when it appeared short after ref. [5], where the authors apply the same technique for massive quenches.

We strongly disagree with our approach being “rather narrow and pedestrian", for instance ref. [5] has more than a thousands of citations, underlying the appreciation of such an approach in the community and the power of the technique.
Indeed, first of all, as it was the case for the solution of massive quenches in LLs, the path integral formulation in imaginary time brings about a much deeper theoretical understanding of the problem. Moreover the elegant solution we provide based on the electrostatic analogy is all but trivial: there are several subtle points first in defining the correct path integral to be solved, and then, in the solution itself which proceeds via some non trivial limits and analytic continuations, which were far from providing in an obvious a priori way the correct result.
Our big effort to make crystal clear all the cumbersome details (another strength of our work acknowledged by the referee) should not be confused with pedestrianism. We want to emphasise that this clarity is not due to the triviality of the problem, but rather to our endeavour towards an in-depth understanding of it.

Finally, as we discuss in the conclusions and as (again) pointed out in the report itself, our method opens up to many possible applications, which we are currently investigating and will be subject of future works. We can add some more details about extensions (e.g. to finite systems), though they are very straightforward, as discussed already, and only require the modification of the equilibrium propagator while leaving the rest of the treatment unchanged.
Apart that, as far as this paper is concerned, we find it complete and self-contained as it is: it is indeed conceived with the specific intent to present a new method which on one side connects to new literature on inhomogeneous gaussian free field, and on the other draws connections to other problems, as the massive quenches and 1d systems coupled to leads.

Altogether, it seems evident to us that the report mainly highlights the strengths of the paper, while (quite surprisingly to us!) ending with a negative outcome. After minor modifications that we are going to include, we therefore ask the referee to reconsider the assessment of the quality of our work and its decision about its publication.

Best regards,

the authors

---

## Round 2 · Referee Report · Anonymous (Referee 2) · 2022-4-21

Report

The authors study the problem of a Luttinger liquid hamiltonian quenched in the Luttinger parameter from $K_0\to K$, exploiting the analogy with a 2D electrostatic problem. In some sense, finding the propagator during this quench problem is analogous to find it for an inhomogeneous Luttinger liquid at equilibrium considered in Ref. 9-13 , after the exchange of space and euclidean time. Since the Luttinger parameter has a simple discontinuity and it jumps from a constant $K_0$ to another constant $K$, the authors surprisingly managed to find a non-trivial analytical solution of the problem. This solution agrees with known results for the massless quench problem in real time (eq (29)) and reproduces the formula for the conductivity in the problem of two Luttinger liquid leads (sec. 6).
The paper is robust, well-written and scientifically sound. It elaborates on the field of Tomonaga Luttinger liquids by interpolating old ideas on quantum quenches (ref. 5-6) with renewed ideas on the electrostatic analogy, recently used in the context of inhomogeneous Luttinger liquids.

I have only a couple of questions on the manuscript. The first is more technical and it regards the additive constant $C$ appearing in the expression of the propagator in eq (13). Is there a physical interpretation of this constant in the electrostatic analogy (something along the line of a diverging background energy)?

The second is more a curiosity: do the authors foresee a possible extension of this calculation to include also simple spatial inhomogeneities?

Concluding, I believe that the manuscript meets the acceptance criteria of SciPost Physics and therefore I recommend it for publication.

  • validity: top
  • significance: high
  • originality: high
  • clarity: high
  • formatting: excellent
  • grammar: excellent

Author:  Paola Ruggiero  on 2022-05-17  [id 2483]

(in reply to Report 2 on 2022-04-21)

We thank the referee for the appreciation of our work. Below we reply to his/her questions.

About the first question: yes, as the referee says the constant can be interpreted as a background energy. Its effect is canceled only for a neutral system.

About adding a spatial inhomogeneity: unfortunately this is not a straightforward generalization. Adding a trap for example would make not only the sound velocity but also the Luttinger parameter space-dependent, thus heavily complicating the problem. We hope we will be able to do something in this direction in the future.

---

## Round 3 · Referee Report · Anonymous (Referee 3) · 2022-5-27

Report

I thank the authors for their detailed response. Unfortunately they appear to have misunderstood my main point, which was not that the whole approach using imaginary time is narrow and pedestrian, but that this particular contribution, although correct, is so, and is not at the level suitable for publication in SciPost.

---

## Round 3 · Referee Report · Anonymous (Referee 2) · 2022-6-5

Report

I thank the authors for their reply to my questions about the manuscript. As stated in my previous report, the manuscript is scientifically correct and well-written. It elaborates on the study of the massless quench in Luttinger liquids, which has a broad interest for the community working on the non-equilibuim dynamics after a quench. On a more technical side, it gives an exact analytic solution for the Green's function in an inhomogeneous medium. Despite the simple form of the inhomogeneity of the Luttinger parameter, this result is quite remarkable. For these reasons, I recommend this manuscript for the publication in SciPost Physics.

---

## Round 3 · Referee Report · Anonymous (Referee 1) · 2022-8-30

Strengths

The paper fills a gap compared to previous works, by proposing to use the imaginary-time path integral for massless quenches in Luttinger Liquids
It makes clear the analogy with the spatial inhomogeneous Luttinger liquid, obtained by a permutation of space and time coordinates.

Weaknesses

The paper recovers an already know result (Eq.29) obtained in Refs. 29,50-52. It is not so clear to guess whether the different method exposed here could give access to other unsolved problems.

Report

The paper is based on the path integral formulation for massless quenches in Luttinger liquids. The main interest of the paper resides into an alternative method to those used in Refs. 29,50-52 to recover the same result. The starting formal expression that allows to switch from real to imaginary time , given by Eq. 4, corresponds to Eq.2 in Ref.5 for instance, where it was developed for massive quenches. Its limitations was addressed in Ref. 6, page 4: "Unfortunately, in confined geometries, only a few field theories with specific boundary conditions can be solved analytically in such a way to have results that can be continued from real to complex values". If I understand well, the present situation corresponds to such an example, thus could be valuable in this respect (maybe the authors could confirm or not this point?). Nonetheless, I have hesitated as it is not clear whether the present method could solve other problems such as those mentioned in the conclusion. The pedagogical value of the paper incites me to recommend its publication in SciPost. The connexion to spatially inhomogeneous Luttinger liquid is also interesting.

Requested changes

I have some remarks about the way similarity to the spatially inhomogeneous Luttinger liquid is presented. Either in the introduction or section 6, authors insist only on the motivation and result for the dc conductance, while the finite frequency Green's function, related to the non-local conductivity, has been obtained (first) in Ref. 38 (and not in Ref.39). The present equation 39 corresponds precisely to Eq.9 in Ref.38. It would be interesting to provide the interpretation thus given of this equation, due to multiple reflexions inside the wire which acts a a Fabry-Perot resonator. By developing the scattering approach to plasmons, the ac conductance was related exactly to the transmission coefficient constructed through evolution of plasmonic modes. As the sum of all transmitted spikes sums up to one (and the sum of reflected ones sum up to zero), the dc conductance is equal to e^2/h$. By the way, it is stated in the introduction (page 2) that "as soon as we consider an inhomogeneous Luttinger parameter K(y,z)... the propagator has to be determined numerically". This is not clearly the case in Refs.38,53,54 which have provided an analytical solution for the propagator.

---

## Round 3 · Author Response

Dear Editors,

We are submitting a revised version of the manuscript
“Electrostatic solution of massless quenches in Luttinger liquids”.

We would like to thank the editors for their work and the referees for
their useful comments and suggestions.

We believe that we thoroughly addressed all the comments of the referees
in the new version of the manuscript.

Sincerely,

Paola Ruggiero
Pasquale Calabrese
Thierry Giamarchi
Laura Foini

---

## Round 3 · List of Changes

- Few typos corrected
- New section added (Section 7)
- Last sentence in Appendix added

---

## Editorial Decision

published